# The BRCAness Landscape of Cancer

**DOI:** 10.3390/cells11233877

**Published:** 2022-12-01

**Authors:** Maoni Guo, San Ming Wang

**Affiliations:** MoE Frontiers Science Center for Precision Oncology, Cancer Centre and Institute of Translational Medicine, Faculty of Health Sciences, University of Macau, Macau 999078, China

**Keywords:** *BRCA1/2*, BRCAness, genetic defects, synthetic lethal, PARP inhibitors

## Abstract

BRCAness refers to the damaged homologous recombination (HR) function due to the defects in HR-involved non-*BRCA1/2* genes. BRCAness is the important marker for the use of synthetic lethal-based PARP inhibitor therapy in breast and ovarian cancer treatment. The success provides an opportunity of applying PARP inhibitor therapy to treat other cancer types with BRCAness features. However, systematic knowledge is lack for BRCAness in different cancer types beyond breast and ovarian cancer. We performed a comprehensive characterization for 40 BRCAness-related genes in 33 cancer types with over 10,000 cancer cases, including pathogenic variation, homozygotic deletion, promoter hypermethylation, gene expression, and clinical correlation of BRCAness in each cancer type. Using *BRCA1*/*BRCA2* mutated breast and ovarian cancer as the control, we observed that BRCAness is widely present in multiple cancer types. Based on the sum of the BRCAneass features in each cancer type, we identified the following 21 cancer types as the potential targets for PARPi therapy: adrenocortical carcinoma, bladder urothelial carcinoma, brain lower grade glioma, colon adenocarcinoma, esophageal carcinoma, head and neck squamous carcinoma, kidney chromophobe, kidney renal clear cell carcinoma, kidney renal papillary cell carcinoma, liver hepatocellular carcinoma, lung adenocarcinoma, lung squamous cell carcinoma, mesothelioma, rectum adenocarcinoma, pancreatic adenocarcinoma, prostate adenocarcinoma, sarcoma, skin cutaneous melanoma, stomach adenocarcinoma, uterine carcinosarcoma, and uterine corpus endometrial carcinoma.

## 1. Introduction

*BRCA1* and *BRCA2* (*BRCA1/2*) play essential roles in repairing double-strand DNA breaks through homologous recombination (HR) [1,2]. Pathogenic variants in *BRCA1/2* damage their function, leads to genome instability, and increased the risk of breast and ovarian cancer [3,4,5,6]. The defected *BRCA1/2* is the specific marker for the use of a synthetic lethal-based PARPi (poly ADP ribose polymerase inhibitors) therapy in cancer treatment. In the process, the PARP inhibitors block PARP function of repairing single-strand breaks and cause the formation of double-strand break upon replication, which cannot be repaired by homologous recombination due to the defected *BRCA1/2* function leading to the death of cancer cells [7,8,9,10,11]. *BRCA1/2* defects are present in about 5% of breast cancer and 20% of ovarian cancer patients, which are the major beneficiaries for the PARP inhibitor therapy [12,13,14].

Homologous recombination pathway involves multiple genes besides *BRCA1/2*. Further, many genes not in HR pathway can also directly or indirectly be involved in HR [15]. In principle, defects in these non-*BRCA1/2* genes could also result in the same consequences caused by the detected *BRCA1/2*. This is described as BRCAness [16]. Based on the studies in breast and ovarian cancer, BRCAness are present in about 20% of breast cancer and 45% of ovarian cancer [17,18], such as the mutations in *RAD51C*, *NBS1*, *BRIP1*, and *PALB2* [19]. Cancer with defected BRCAness shared many features such as the cancer with defected *BRCA1/2*. For instance, BRCAness cancer genome had high abundant C-to-T transitions [20], shared the “Signature 3” of mutation signatures in cancer [21,22], and contained high level of promoter hypermethylation [23], prone to triple-negative breast cancer (TNBC) and “Basal” subtypes, and benefited from PARPi treatment [24,25,26,27,28].

The success of PARPi therapy in *BRCA1/2* defected breast and ovarian cancer attracts great interests in exploring its potential to treat other cancer types with BRCAness features. Indeed, it was observed that BRCAness was present in prostate cancer, colon cancer and pancreatic cancer [29,30,31], PARP inhibitor treatment improved survival of pancreatic cancer carrying mutated HRR deficient-related genes of *ATM*, *BRCA1/2*, *CHEK2*, *PALB2*, *RAD51C*, *RAD51D* [32], and enhanced treatment response in metastatic prostate cancer carrying mutated HRR deficient-related genes *ATM*, *BRCA1/2*, *BRIP1*, *BARD1*, *CDK12*, *CHEK1*, *CHEK2*, *FANCL*, *PALB2*, *RAD5C* [33,34,35]. However, systematic knowledge is lack for BRCAness across cancer type spectrum. This largely restricts the use of synthetic lethal-based PARPi therapy as a universal option in cancer treatment.

We hypothesized that BRCAness could be a common phenomenon in cancer. To test our hypothesis, we performed a BRCAness characterization in 33 cancer types, covering pathogenic variation, homozygotic deletion, promoter hypermethylation and expression, copy number variation, and clinical correlation for BRCAness-related genes. By using *BRCA1/2*-mutated breast and ovarian cancer as the references, we observed the wide presence of BRCAness signatures in multiple cancer types. Our study provides a foundation to further test the potential of using PARPi therapy to treat the cancer types with BRCAness features.

## 2. Materials and Methods

### 2.1. Sources of Genome Data across 33 Cancer Types

We identified 40 BRCAness genes from literatures (*ATM*, *ATR*, *AURKA*, *BAP1*, *BARD1*, *BLM*, *BRCA1*, *BRCA2*, *BRIP1*, *CDK12*, *CHD4*, *CHEK1*, *CHEK2*, *EMSY*, *ERCC1*, *FANCA*, *FANCC*, *FANCD2*, *FANCE*, *FANCF*, *FANCI*, *KMT2A*, *MRE11A*, *MYC*, *NBS1*, *PALB2*, *PARP1*, *PAXIP1*, *PLK1*, *PTEN*, *RAD50*, *RAD51*, *RAD51B*, *RAD51C*, *RAD51D*, *RAD52*, *SAMHD1*, *SEM1*, *TP53*, *TP53BP1*, *WEE1*, *WRN*) [16,36,37]. We collected the BRCAness genomic and clinical information from these two resources: UCSC xena (http://xena.ucsc.edu/, accessed on 18 November 2020) and PanCanAtlas (https://gdc.cancer.gov/node/905/, accessed on 20 November 2020) covering 33 cancer types [38]. The details of data information are as follows: Variation data from over 10,000 cancer patients were from TCGA MAF file in PanCanAtlas; copy number variation (CNV) data detected by Affymetrix SNP 6.0 arrays were from UCSC xena; DNA methylation data detected by Illumina HumanMethylation450 BeadChip platform were from PanCanAtlas; RNA-seq data with normalized batch effects and log2 (norm_value+1) gene expression for all 33 cancer types were from PanCanAtlas; Clinical survival data were from UCSC xena. GISTIC2 was used to identify the genomic regions with significant gain or loss [39].

### 2.2. Pathogenic Variant Data in BRCAness Genes

To identify pathogenic and likely pathogenic variants (PLPs) for each BRCAness gene across each cancer type, we first extracted the variants that passed filtering and belonged to non-silent subtypes from TCGA pan-cancer variation data. We further downloaded all variations from ClinVar (https://www.ncbi.nlm.nih.gov/clinvar/, accessed on 25 January 2021) and COSMIC (the Catalogue of Somatic Mutations in Cancer, https://cancer.sanger.ac.uk/cosmic, accessed on 25 January 2021) databases and extracted the “Pathogenic” and “Likely Pathogenic” variants from ClinVar and “Pathogenic” variants from COSMIC (PLP). We compared the TCGA variants to ClinVar/COSMIC PLP variants to identify germline and somatic PLPs for the BRCAness genes in each cancer type. We calculated variation frequency for the PLP variants in each BRCAness gene in each cancer type and tested the correlation of PLP variation frequencies between BRCAness genes and *BRCA1/2* in all 33 cancer types by Pearson’s correlation coefficient analysis. *p*-value < 0.05 was considered as statistically significant.

### 2.3. CNV Data in BRCAness Genes

We used GISTIC2 to identify CNV at gene-level [39], with gene count of “−2” defined as homozygotic mutation. We retained homozygous deletion for CNV analysis across all 33 cancer types. We calculated homozygous deletion frequency for each BRCAness gene in each cancer type, and clustered homozygous deletion frequencies for all 33 cancer types using R package “ComplexHeatmap” [40]. We also tested the correlation of frequencies for homozygous deletions between *BRCA1/2* and BRCAness genes in all 33 cancer types by Pearson’s correlation coefficient analysis. *p*-value < 0.05 was used as statistically significant in correlation analyses.

### 2.4. DNA Methylation Data in BRCAness Genes

We used DNA methylation data from the HumanMethylation450 arrays. We mapped the probes to each BRCAness genes using the Illumina GPL13534 platform (https://www.ncbi.nlm.nih.gov/geo/query/acc.cgi?acc=GPL13534, accessed on 23 May 2021). Promoter for *BRCA1/2* and BRCAness genes was defined as within 10 kb surrounding transcription start site (TSS) for each gene. We obtained hypermethylated data with mean gene-level promoter methylation values > 0.3. We clustered promoter methylation levels (mean promoter methylation values) for *BRCA1/2* and BRCAness genes in 33 cancer types using R package “ComplexHeatmap” [40].

### 2.5. Gene Expression and Functional Enrichment Analysis

We used the Wilcox’s rank sum test to identify differentially expressed genes (DEGs) in each cancer type. We adjusted the p-values by Benjamini–Hochberg (BH) method [41]. We defined the differentially expressed genes with adjusted *p*-values < 0.01 and at least two-fold changes in expression level. We also performed Gene Ontology (GO) and Kyoto Encyclopedia of Genes and Genomes (KEGG) functional enrichment annotation using the DAVID tool (http://david.abcc.ncifcrf.gov/, accessed on 23 May 2021, version 6.8), with Benjamini p.adjust < 0.05 as statistically significance [42].

### 2.6. Clinical Relevance of BRCAness Genes

We divided the patients into high-risk and low-risk groups based on the median expression of each BRCAness gene. We used the Kaplan–Meier method to generate survival curves, and two-sided log-rank tests to assess the differences in overall survival between the high-risk and low-risk groups by using the R package “survival” and *p*-value < 0.05 as significant difference. We calculated the hazard ratio and 95% confidence level and plotted the results by R “forestplot” package.

### 2.7. Identifying Candidate Cancer Types for PARPi Therapy

Taking breast cancer and ovarian cancer as the references, we ranked the 33 cancer types based on following six integrated features: for somatic and germline pathogenic variation, and homozygous deletion, we calculated the frequencies of all BRCAness genes in each cancer type; for promoter methylation, we calculated the total promoter methylation level of all BRCAness genes in each cancer type; for gene expression, we calculated the total number of differentially expressed BRCAness genes in each cancer type; for prognostics, we located the number of risky BRCAness genes in each cancer type. We set BRCA/OV = 1 to identify BRCAness cancer types that only the cancer type with >=1 qualified as BRCAness cancer types.

## 3. Results

### 3.1. Overview of the Study

Functional enrichment analysis showed that the 40 BRCAness genes were involved in homologous recombination, DNA repair, cell cycle, and Fanconi anemia pathways (Figure 1A). Using the TCGA multi-omics data from 33 cancer types derived from over 10,000 cancer cases, we characterized BRCAness genes and their clinical relevance (Figure 1B, Appendix A) by focusing on six characters: pathogenic variation, CNV, homozygosity, DNA methylation and gene expression, and clinical prognosis. In each analysis, *BRCA1/2* status was used as gene-level controls, and *BRCA1/2* mutated-breast and ovarian cancer were used as the cancer type-level controls.

### 3.2. Pathogenic Variation in BRCAness Genes in Different Cancer Types

Pathogenic variation in BRCAness gene is the direct indication for the presence of BRCAness in cancer. We searched for both germline and somatic pathogenic variants in *BRCA1/2* and BRCAness genes in each of the 33 cancer types and identified a total of 808 germline and 4017 somatic pathogenic mutations in *BRCA1, BRCA2* and 37 of the 38 BRCAness genes distributed in 33 cancer types. On average, there were 24 germline and 122 somatic mutations per cancer type distributed at different frequencies in different cancer types (Appendix A). As expected, *BRCA1/2* had both germline and somatic pathogenic variation in breast and ovarian cancer, whereas BRCAness genes had high prevalence of somatic pathogenic variation distributed in cancer type-specific manners (Figure 2, Appendix A). The prevalence of somatic pathogenic variation in BRCAness genes ranged between 0.14% and 39.81%, with TP53 as the highest among all BRCAness genes, followed by *PTEN*, *ATM*, *CHD4*, *KMT2A*, *ATR*, *CDK12*, *BAP1* and *TP53BP1* higher than *BRCA2*, and *FANCD2*, *RAD50*, *MYC*, *BRIP1*, *FANCI*, *PLK1*, *PAXIP1*, *CHEK2*, *PARP1* and *SAMHD1*, all of which were higher than *BRCA1* (Figure 2, Appendix A). Of the 33 cancer types, 6 had higher prevalence of BRCAness pathogenic variation than breast cancer (BRCA) and 19 higher than ovarian cancer (OV), with UCEC as the highest of 1787 among all 33 cancer types (Figure 2, Appendix A). LAML, TGCT, PCPG, UVM, CHOL, and THYM had very low prevalence of BRCAness somatic and germline pathogenic variation. For example, LAML (acute myeloid leukemia) had neither somatic nor germline pathogenic variation in BRCAness genes, although AML has consistent cytogenetic abnormality (Figure 2, Appendix A). Therefore, these cancer types were unlikely relevant with BRCAness. Then, we performed the variation frequency correlation analysis to explore whether similar variant patterns exist between BRCAness genes and BRCA1/2. We performed Pearson correlation analysis to test the correlation between BRCAness pathogenic variation and *BRCA1/2* genes. Except *TP53*, *BAP1*, *PALB2*, *ERCC1* with the somatic pathogenic variation, nearly all BRCAness genes had significant correlation with *BRCA1* and/or *BRCA2* (Figure 3A,B), and about a third of BRCAness genes with germline pathogenic variation had significant correlation with *BRCA1* and/or *BRCA2* (Figure 3C,D).

### 3.3. Homozygotic Deletion Patterns between BRCA1/2 and BRCAness Genes in Different Cancer Types

Pathogenic variation in human BRCA is predominantly heterozygotic due to the embryo lethal effects of *BRCA1/2* homozygotic variation. Therefore, homozygosity provides an important indicator to test the similarity between BRCAness and *BRCA1/2* pathogenic variation. From the homozygotic deletion identified in BRCAness genes, we observed low frequency of 0.01–1.16% homozygotic deletion in BRCAness genes, comparing to 0.16% in *BRCA1* and 0.32% in *BRCA2* (Figure 4A). Of the 40 BRCAness genes, 21 (*RAD51D*, *PARP1*, *WEE1*, *FANCC*, *RAD52*, *CDK12*, *ERCC1*, *CHD4*, *NBS1*, *PLK1*, *FANCF*, *ATR*, *RAD51C*, *FANCE*, *PALB2*, *BRIP1*, *BLM*, *SAMHD1*, *FANCI*, *MYC*, *AURKA*) had lower than the 0.16% in *BRCA1* and 29 (*TP53BP1*, *PAXIP1*, *RAD51B*, *FANCD2*, *MRE11A*, *CHEK2*, *RAD50*, *BARD1*, *RAD51D*, *PARP1*, *WEE1*, *FANCC*, *RAD52*, *CDK12*, *ERCC1*, *CHD4*, *NBS1*, *PLK1*, *FANCF*, *ATR*, *RAD51C*, *FANCE*, *PALB2*, *BRIP1*, *BLM*, *SAMHD1*, *FANCI*, *MYC*, *AURKA*) had lower than the 0.32% in *BRCA2* (Figure 4A). Nine BRCAness genes had significant correlation with *BRCA1* and 15 with *BRCA2* (Figure 4B–E). Of the 33 cancer types, 23 (LUAD, KIRC, STAD, ESCA, UVM, TGCT, ACC, COAD, LIHC, HNSC, UCEC, LGG, GBM, UCS, LAML, READ, KIRP, THYM, PAAD, THCA, PCPG, KICH, CHOL) were higher than BRCA cancer and OV cancer (Appendix A). The results showed that like BRCA and the related cancer types, homozygotic variation was insignificantly present in BRCAness genes and their related cancer types.

### 3.4. Methylation and Expression Patterns between BRCA1/2 and BRCAness Genes in Different Cancer Types

Promoter methylation plays important roles in de-regulation of *BRCA1/2* expression in breast and ovarian cancer [43]. We investigated promoter methylation in BRCAness genes across all 33 cancer types. By using promoter methylation level in *BRCA2* as the cutoff, 16 of the 40 BRCAness gene promoters were hypermethylated across nearly all cancer types (Figure 5A, Appendix A). We then tested the effects of promoter hypermethylation on BRCAness gene expression. In the 19 cancer types with expression data available from at least 5 normal samples as control, 10 cancer types of PRAD, READ, KIRP, COAD, STAD, LUAD, THCA, LIHC, ESCA, PAAD had lower expression of BRCAness genes than their normal controls by referring to breast cancer (ovarian cancer was not included due to the lack of expression data from normal ovarian control) (Figure 5B). Of the 33 cancer types, BRCA cancer had total levels of *BRCA1/2* and BRCAness gene promoter hypermethylation of 14.244 and OV cancer had 13.926. By using OV cancer as the cutoff, 20 of the 33 cancer types had promoter hypermethylation (Appendix A). The hypermethylation in OV had distinct features from other cancer types: a group of BRCAness genes of *MER11A*, *RAD51*, *PALB2*, *CHD4*, *BRIP1*, *MYC*, and *FANCE* were increasingly expressed whereas another group of *WRN*, *CDK12*, *CHEK2*, *FANCA*, and *TP53* was decreasingly expressed (Figure 5C). Each cancer type also had specific hypermethylated BRCAness genes (Figure 5C). For example, *FANCF* promoter was hypermethylated and had decreased expression in eight cancer types of BLCA, CESC, COAD, DLBC, ESCA, HNSC, UCEC, and UCS (Figure 5C). The results highlighted that *BRCA1/2* and BRCAness genes shared high similarity of promoter hypermethylation across multiple cancer types.

### 3.5. Prognostics of BRCAness Gene Expression across Different Cancer Types

We analyzed the prognostic potential of altered BRCAness gene expression across different cancer types. We observed that the altered expression of each BRCAness gene was significantly correlated with the overall survival in at least one cancer type except TGCT, THCA, and UCS (Figure 6A). For example, increased expression of *AURKA* was correlated with worse survival in 16 of the 33 cancer types including KIRP (log-rank *p* = 3.41 × 10^−8^, KICH (log-rank *p* = 5.20 × 10^−6^), and KIRC (log-rank *p* = 1.11 × 10^−10^) (Figure 6B); increased expression of *PTEN* was correlated with better survival in 6 cancer types including LGG (log-rank *p* = 4.44 × 10^−9^), KIRC (log-rank *p* = 4.21 × 10^−3^), LIHC (log-rank *p* = 2.64 × 10^−2^), UCEC (log-rank *p* = 4.25 × 10^−5^), PRAD (log-rank *p* = 7.07 × 10^−2^) and LAML (log-rank *p* = 3.64 × 10^−2^) (Figure 6B). Altered expression of BRCAness genes can classify cancer patients into high- and low-risk groups. For example, based on altered expression of BRCAness genes, KIRC patients were divided into the lower expression group of 112 patients and the higher expression group of 422 patients (Figure 6C), in which the lower expression subgroup had significantly better survival than the higher expression subgroup (Figure 6D and Appendix A). The results showed that similar to *BRCA1/2* in breast and ovarian cancer, BRCAness genes were the prognostic markers for multiple cancer types.

### 3.6. BRCAness Cancer Types Sharing High Similarity with BRCA and OV Cancer

Data from above analyses identified multiple cancer types enriched with BRCAness features. We ranked the 33 cancer types based on the sum of BRCAneass features in the six features of somatic pathogenic variation, germline pathogenic variation, homozygous deletion, expression, and clinical prognosis. By using the sum in both BRCA cancer and OV cancer values = 1 as the cut-off, we observed that the following 21 cancer types had BRCAness features higher than BRCA cancer and OV cancer: UCEC, BLCA, PAAD, LGG, SARC, LUAD, KICH, UCS, ACC, COAD, LIHC, ESCA, HNSC, READ, STAD, SKCM, LUSC, MESO, KIRC, KIRP, and PRAD (Table 1). Of the 21 cancer types, UCEC (uterine corpus endometrial carcinoma) was the highest as referred by BRCA cancer and OV cancer.

## 4. Discussion

By characterizing pathogenic variants, somatic variants, homozygotic deletion, promoter methylation, gene expression, and clinical prognosis, our study portraits a comprehensive view for BRCAness landscape in cancer and reveals that BRCAness is widely present in different cancer types. We observed that genetic variation was widely presence in BRCAness genes in multiple cancer types, homozygotic variation was rare event in BRCAness genes as in BRCA, promoter methylation was common in BRCAness genes and caused alterative expression, the defects in BRCAness genes were strong prognostics markers as BRCA. By referring to the sum of BRCAness features higher than BRCA defected breast and ovarian cancer, we identified 21 BRCAness cancer types as the candidate targets for PARPi trial to further determine the efficacy of PARPi therapy in each cancer type.

By targeting multiple oncogenic components, synthetic lethal has shown promising potential as best exemplified by using PARPi to treat BRCA1/2 mutated breast and ovarian cancer. In our current study, we analyzed the potential of using PARPi therapy to treat other cancer types with BRCAness features. Through analyzing multiple features in 33 cancer types, our study provided the following evidence showing high similarity between BRCAness and *BRCA1/2* mutation in multiple cancer types: (1) Genetic variation was widely present in BRCAness genes in multiple cancer types as represented by UCEC, BLCA, LUSC, HNSC, STAD, and COAD [44]; (2) homozygotic variation was a rare event in BRCAness genes as in *BRCA1/2* mutation. Similar to the embryonic lethal effects in *BRCA1/2*, homozygous variation in BRCAness genes was not common across different cancer types although homozygous deletions in some BRCAness genes (*FANCD2*, *PTEN*, *TP53*, *BRCA2*) could be present [45]; (3) promoter methylation was common in many BRCAness genes and caused alterative expression [46]. Similar to the promoter methylation occurred in *BRCA1/2*, promoter methylation was present in nearly half of the BRCAness genes, and their expression were silenced in many cancer types in cancer type-specific manner; (4) the defects in BRCAness genes were strong prognostics markers as *BRCA1/2* mutation, as shown in the BRCAness genes of *CHEK2*, *ATM*, *RAD51D*, *EMSY*, *PALB2*, *BRIP1*, *ERCC1*, *RAD50*, *ATR*, *RAD51C* in ovarian cancer [47].

Our study reveals that BRCAness defects are commonly present in multiple cancer types as *BRCA1/2* defects in breast and ovarian cancer. Therefore, it opens a possibility to further test the potential of expanding PARPi therapy from breast and ovarian cancer to more cancer types with BRCAness features.

## Figures and Tables

**Figure 1 cells-11-03877-f001:**
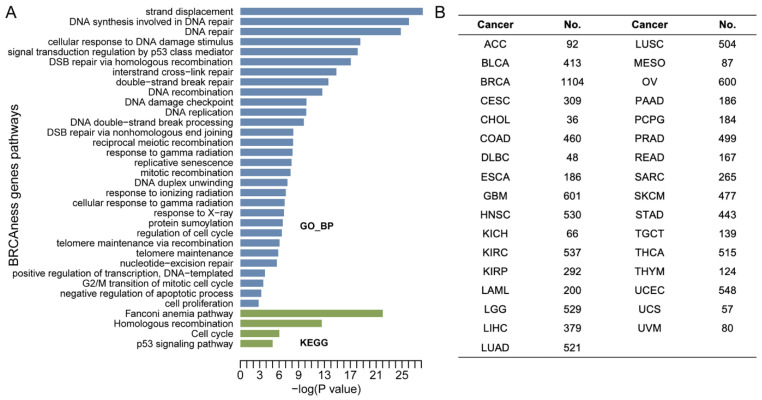
Scheme of the analysis. (**A**) Functional enrichment analysis for BRCAness genes. It shows the barplot for the top 30 significant GO functional enriched biological processes (BPs), and KEGG pathways. (**B**) The number of patients included each cancer type.

**Figure 2 cells-11-03877-f002:**
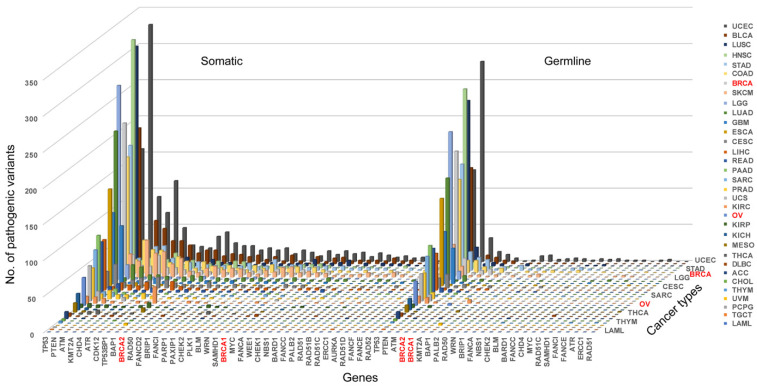
Distribution of pathogenic variation of BRCAness genes in different cancer types. Left part shows the distribution of somatic and right part shows the germline pathogenic variants in BRCAness genes and cancer types. BRCA1 and BRCA2 in red show their corresponding distribution for comparison with BRCAness genes. Abbreviations: ACC, adrenocortical carcinoma; BLCA, bladder urothelial carcinoma; BRCA, breast invasive carcinoma; CESC, cervical squamous cell carcinoma and endocervical adenocarcinoma; CHOL, cholangiocarcinoma; COAD, colon adenocarcinoma; DLBC, lymphoid neoplasm diffuse large B-cell lymphoma; ESCA, esophageal carcinoma; GBM, glioblastoma multiforme; HNSC, head and neck squamous carcinoma; KICH, kidney chromophobe; KIRC, kidney renal clear cell carcinoma; KIRP, kidney renal papillary cell carcinoma; LAML, acute myeloid leukemia; LGG, brain lower grade glioma; LIHC, liver hepatocellular carcinoma; LUAD, lung adenocarcinoma; LUSC, lung squamous cell carcinoma; MESO, mesothelioma; OV, ovarian serous cystadenocarcinoma; PAAD, pancreatic adenocarcinoma; PCPG, pheochromocytoma and paraganglioma; PRAD, prostate adenocarcinoma; READ, rectum adenocarcinoma; SARC, sarcoma; SKCM, skin cutaneous melanoma; STAD, stomach adenocarcinoma; TGCT, testicular germ cell tumors; THCA, thyroid carcinoma; THYM, thymoma; UCEC, uterine corpus endometrial carcinoma; UCS, uterine carcinosarcoma; UVM, uveal melanoma.

**Figure 3 cells-11-03877-f003:**
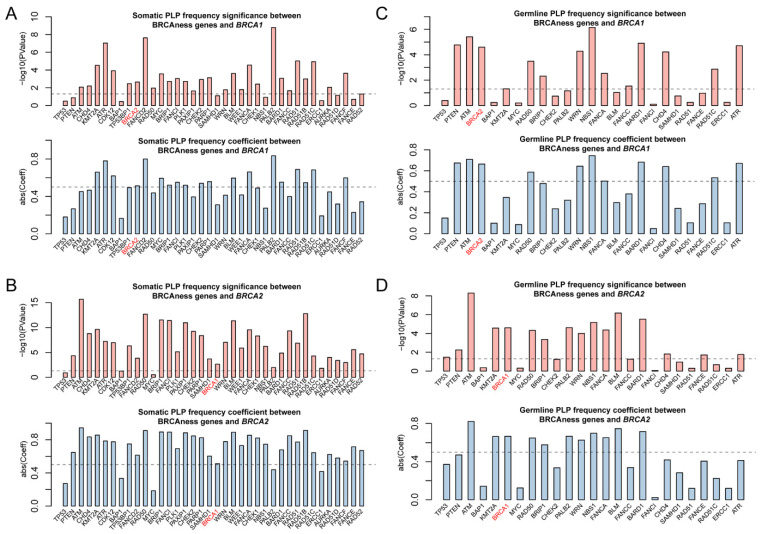
Correlation of pathogenic variation of BRCAness genes across 33 cancer types. (**A**,**B**) Significant correlation for the frequencies of somatic and germline pathogenic variants between individual BRCAness gene and *BRCA1*. (**C**,**D**). Significant correlation for the frequencies of somatic and germline pathogenic variants between individual BRCAness gene and *BRCA2*. The pairs with *p*-value < 0.05 were regarded as significant. The pairs with an absolute coefficient > 0.5 were regarded as significantly correlated.

**Figure 4 cells-11-03877-f004:**
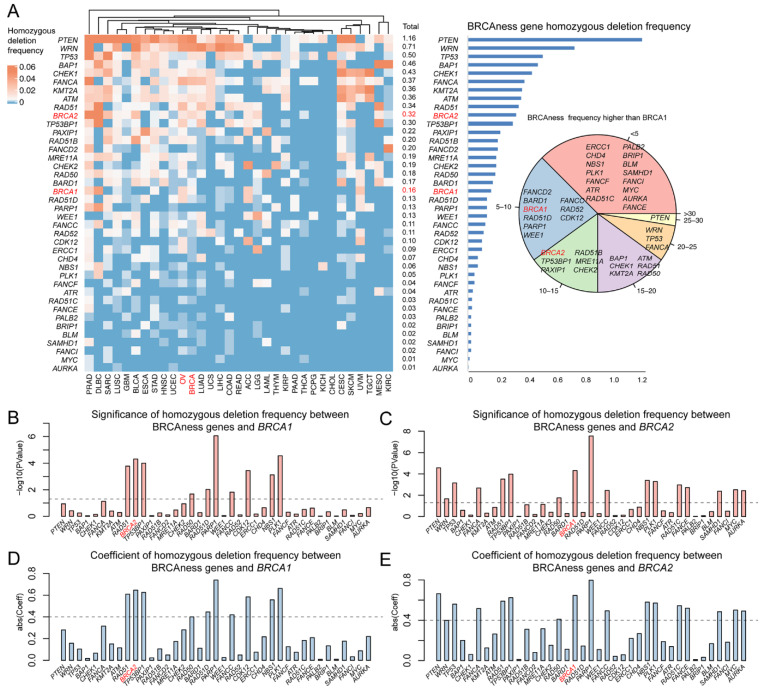
Homozygotic deletions of BRCAness genes in different cancer types. (**A**) The frequency of homozygotic deletion in each BRCAness gene across 33 cancer types (left) and in all cancer types (right). In the inset on right, we displayed gene list falling into each classification. For example, the genes list (*FANCD2*, *BARD1*, *FANCC*, *BRCA1*, *RAD52*, *RAD51D*, *CDK12*, *PARP1*, *WEE1*) in the blue region represent these genes have higher homozygotic variation frequency than *BRCA1* in 5 to 10 cancer types. (**B**,**C**) The significance of the homozygotic frequencies between each BRCAness gene and *BRCA1/2*. The pairs with a *p*-value < 0.05 were regarded as significant. (**D**,**E**) The correlation of homozygotic frequencies between BRCAness genes and *BRCA1/2*. The pairs with an absolute coefficient > 0.4 were regarded as significantly correlated.

**Figure 5 cells-11-03877-f005:**
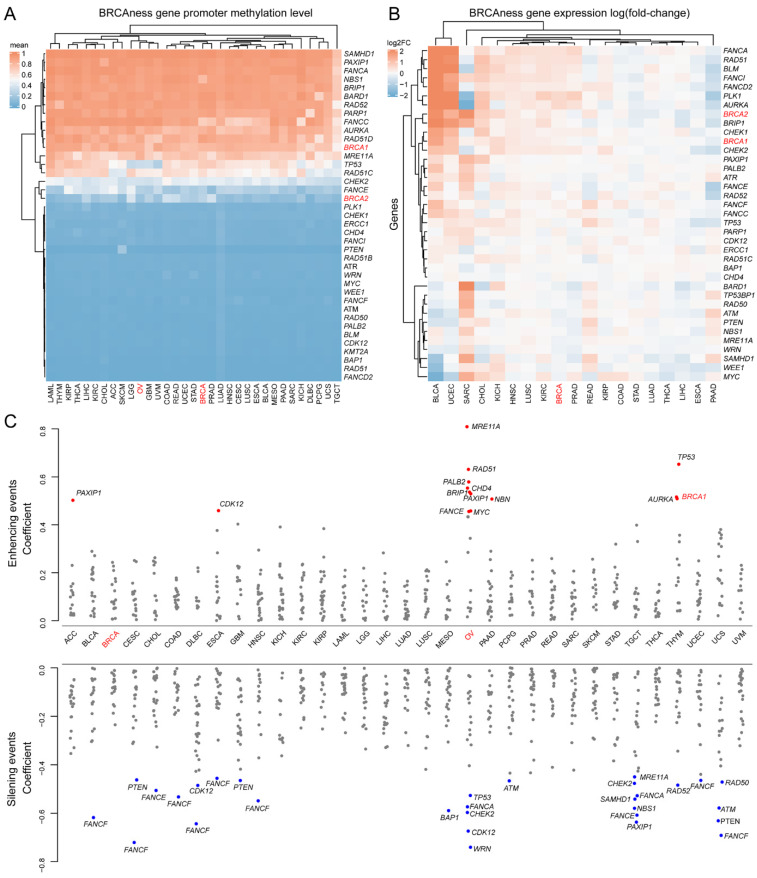
Methylation and transcriptional regulation of BRCAness genes. (**A**) Promoter methylation of BRCAness genes across 33 cancer types. It showed the mean beta value of all promoter methylation sites. Red: hypermethylated genes; blue: hypomethylated genes. By using *BRCA2* as the cutoff, 18 BRCAness genes were hypermethylated and 24 were hypomethylated. (**B**) Altered expression of BRCAness genes in 19 cancer types with available expression data from normal controls. Red: up-regulated genes; blue: down-regulated genes. (**C**) Cancer type-specific silencing effects of promoter methylation in BRCAness genes. It showed that promoter methylation silenced the expression in the majority of BRCAness genes, except the enhanced expression in 13 BRCAness genes, and the silencing effects were highly cancer type specific. Blue: silencing; red: enhancing. Pearson’s correlation coefficient > 0.5 represents significant associations.

**Figure 6 cells-11-03877-f006:**
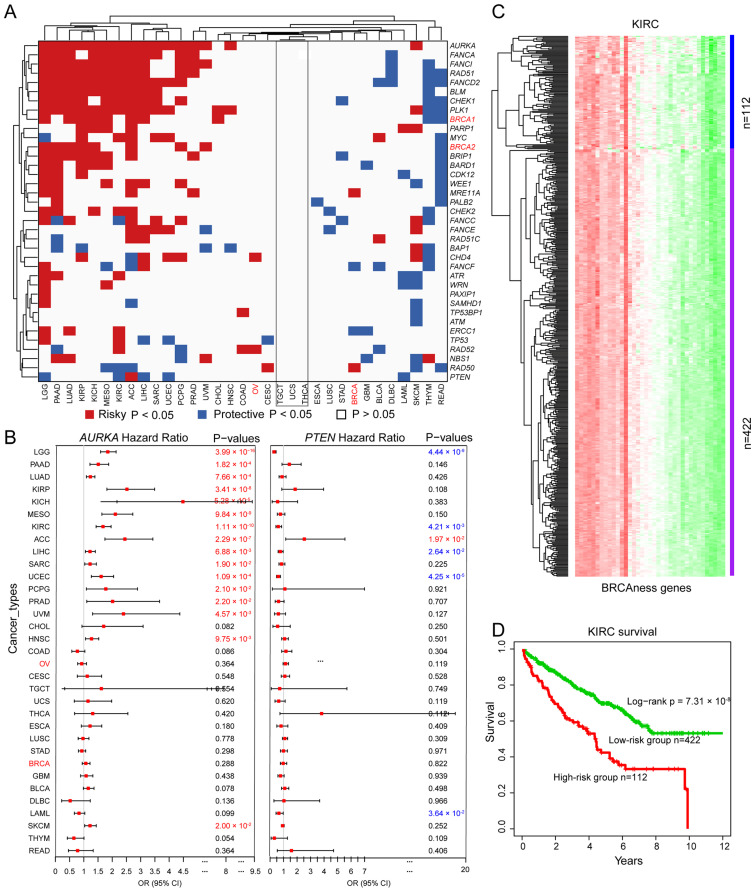
Clinical relevance of BRCAness genes across cancer types. (**A**) Correlation between expression of BRCAness genes and patient survival. OR, odds ratio; CI: confidence interval; Red *p*-values represents a higher expression of BRCAness genes associated with worse survival, and blue *p*-values represents an association with better survival. Only *p* values < 0.05 are shown. (**B**) Distribution of hazard ratios across different cancer types in *AURKA* and *PTEN*. Red: the cancer types for those the expression levels of *AURKA* or *PTEN* are risky factors; Blue: the cancer types for those the expression levels of *PTEN* are protective factors. (**C**) Heatmap showing the clustered expression of all BRCAness genes in KIRC cancer patients. Red, higher levels of gene expression; Green, lower levels of gene expression. (**D**) Kaplan–Meier survival plot of KIRC patients grouped by the altered expression patterns of BRCAness genes.

**Table 1 cells-11-03877-t001:** The 21 candidate cancer types for PARPi test as referred by both BRCA cancer and OV cancer.

Cancer	Rank by BRCA Cancer						Rank by OV Cancer						Sum
	Somatic	Germline	Homozygotic	Methylation	Expression	Prognosis	Somatic	Germline	Homozygotic	Methylation	Expression	Prognosis	
UCEC	5.9	4.7	0.7	1.0	7.0	3.0	11.0	17.2	0.5	1.0	7.0	3.0	62.1
BLCA	3.8	2.0	1.3	1.0	10.5	1.0	4.8	14.6	1.0	1.0	10.5	1.0	52.6
PAAD	1.5	1.9	0.2	1.0	5.5	9.0	4.5	6.1	0.1	1.0	5.5	9.0	45.3
LGG	1.5	1.9	0.7	1.0	-	12.0	4.6	5.7	0.5	1.1	-	12.0	41.0
SARC	1.1	1.2	1.8	1.0	8.5	4.5	2.9	4.0	1.4	1.0	8.5	4.5	40.5
LUAD	1.8	1.5	1.0	1.0	0.0	7.0	3.5	6.9	0.8	1.0	0.0	7.0	31.6
KICH	1.2	1.2	0.1	1.0	3.0	6.5	2.9	4.2	0.1	1.0	3.0	6.5	30.5
UCS	3.2	4.0	0.7	0.9	-	0.0	9.6	10.1	0.5	0.9	-	0.0	30.0
ACC	0.4	0.7	0.8	1.0	-	10.0	1.6	1.4	0.6	1.0	-	10.0	27.4
COAD	2.9	2.2	0.7	1.0	0.0	1.0	5.3	11.0	0.6	1.0	0.0	1.0	26.8
LIHC	1.1	0.8	0.7	1.0	0.5	7.0	1.8	4.4	0.6	1.0	0.5	7.0	26.4
ESCA	2.0	3.3	1.0	1.0	1.0	0.0	7.7	7.7	0.8	1.0	1.0	0.0	26.4
HNSC	2.5	2.4	0.7	1.0	0.5	1.0	5.8	9.5	0.5	1.0	0.5	1.0	26.4
READ	2.3	2.4	0.5	1.0	2.0	0.0	5.8	8.5	0.4	1.0	2.0	0.0	25.9
STAD	2.8	2.6	1.0	1.0	0.0	0.0	6.0	10.5	0.8	1.0	0.0	0.0	25.6
SKCM	2.5	0.8	1.1	1.0	-	3.5	1.9	9.3	0.9	1.0	-	3.5	25.5
LUSC	2.7	2.6	1.1	1.0	0.0	0.0	6.3	9.9	0.8	1.0	0.0	0.0	25.3
MESO	0.7	0.8	1.2	1.0	-	7.0	1.9	2.4	0.9	1.0	-	7.0	23.9
KIRC	0.5	0.3	1.0	1.0	0.5	7.0	0.8	1.8	0.8	1.0	0.5	7.0	22.3
KIRP	0.4	0.1	0.5	1.0	0.0	7.5	0.3	1.5	0.4	1.1	0.0	7.5	20.3
PRAD	0.5	0.5	3.1	1.0	1.0	3.0	1.1	2.0	2.4	1.0	1.0	3.0	19.6
BRCA&OV	1.0	1.0	1.0	1.0	1.0	1.0	1.0	1.0	1.0	1.0	1.0	1.0	12.0

## Data Availability

Not applicable.

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
