# Peer review of "The BRCAness Landscape of Cancer"

_cells, 2022, doi:10.3390/cells11233877_

Round 1

Reviewer 1 Report

Thank you for this assessment of "BRCAness" in cancer types beyong breast and ovarian cancer - an interesting idea for associated treatment implications as described.

Some feedback:

1. There needs to be a careful revision of the manuscript for English language use and typos - eg PPRPi instead of PARPi in paragraph 3.

2. Further in paragraph 3 - description of BRCAness genes includes reference to BRCA1/2. Perhaps these gene lists should be described as "HRR deficient" related genes?

3. Define "BRCA" (as in BRCA versus BRCAness) to encapsulate BRCA1 and BRCA2 mutated (deficient? was methylation in these genes included?) cancers.

4. TP53 is included in you analysis, with a high prevalence of somatic mutations detected ("the highest amongst BRCAness genes"). TP53 is a very commonly mutated gene across many cancer types (~40-50%). In ovarian cancer specifically, up to 96% of all high-grade serous ovarian cancers harbor a somatic pathogenic change in TP53. These changes occur in HRR deficient tumours, and those with intact HRR (and even mutually exclusive genetic events, such as CCNE1 amplification). They are a hallmark marker of HGSOC pathological classification, and not BRCAness. I think TP53 should be excluded from the analysis for this reason. (Also not a target for treatment options). 

5. Was there any more specific analysis of histopathological subtypes when BRCAness was evident? ie across different histopathological subtypes of the same cancer type?

Author Response

Thank you for this assessment of "BRCAness" in cancer types beyond breast and ovarian cancer - an interesting idea for associated treatment implications as described.

Some feedback:

  1. There needs to be a careful revision of the manuscript for English language use and typos - eg PPRPi instead of PARPi in paragraph 3.

Reply:

We have checked though the manuscript and revised the typos.

  1. Further in paragraph 3 - description of BRCAness genes includes reference to BRCA1/2. Perhaps these gene lists should be described as "HRR deficient" related genes?

Reply:

Thank you very much for your suggestion. We replaced “BRCAness genes” with “HRR deficient-related genes.

  1. Define "BRCA" (as in BRCA versus BRCAness) to encapsulate BRCA1 and BRCA2 mutated (deficient? was methylation in these genes included?) cancers.

Reply:

Thank you very much for your comments. We defined “BRCA” (as in BRCA versus BRCAness) to encapsulate BRCA1 and BRCA2 deficient cancers  from not only mutation (including Pathogenic variant, homozygous deletion) but also promoter methylation levels in these genes (see Figures 2,3,4).

  1. TP53 is included in your analysis, with a high prevalence of somatic mutations detected ("the highest amongst BRCAness genes"). TP53 is a very commonly mutated gene across many cancer types (~40-50%). In ovarian cancer specifically, up to 96% of all high-grade serous ovarian cancers harbor a somatic pathogenic change in TP53. These changes occur in HRR deficient tumours, and those with intact HRR (and even mutually exclusive genetic events, such as CCNE1 amplification). They are a hallmark marker of HGSOC pathological classification, and not BRCAness. I think TP53 should be excluded from the analysis for this reason. (Also, not a target for treatment options).

Reply:

Thank you very much for your comments. TP53codes for p53. t regulates cell division by controlling cellular growth and division. p53 is also essential for regulating DNA damage repair. If the damaged DNA can be repaired, p53 will activate other genes to fix the damage; if the damaged DNA cannot be repaired, p53 will signal the cell to undergo apoptosis. By stopping cells with mutated or damaged DNA from dividing, p53 helps prevent the development of tumor. Although it may not be a direct marker of BRCAness, TP53 is an indispensable key modulator in DNA damage repair, plays a supervisory role in preventing tumorigenesis together with other HRR defective genes. Therefore, we consider it necessary to keep TP53  as an indicator for the evaluation of BRCAness.

  1. Was there any more specific analysis of histopathological subtypes when BRCAness was evident? ie across different histopathological subtypes of the same cancer type?

Reply:

Thank you very much for your suggestion. As many original data didn’t provide histopathological information, we didn’t perform specific analysis of histopathological subtypes of the same cancer type. However, this is really a good idea. In our next phase BRCAness study, we would certainly explore this issue.

Reviewer 2 Report

Mutations in BRCA1/2 cause breast cancer, ovarian cancer and other cancers. BRCA1/2 function in homologous recombination (HR) repair in DNA double strand breaks (DSB). Poly(ADP-ribose) polymerase inhibitors (PARPi) block single strand breaks (SSB) repair and lead to DSB, and eventually cause cell death in BRCA1/2 mutated cells, based on this they are used for the treatment of BRCA1/2 mutated cancers. Like BRCA1/2, any genes function in HR, these genes mutated cancers can be treated by PARPi, which is so called BRCAness. Using bioinformatics approaches, in this manuscript authors used 40 BRCAness genes, 33 cancer types, and >10,000 cancers, and concluded that 21 cancer types are suitable for PARPi treatment. Therefore, this study is a useful step to expand the use of PARPi in wide cancer treatment. However, we have to realize it is a hot topic, and many relevant research and review papers have been published.

Here are some issues (from major to minor):

1.     The definition of BRCAness. The first sentence in abstract “BRCAness refers to the damaged homologous recombination (HR) function due to the defects in HR-involved non-BRCA1/2 genes” Reading referred Ref 16 for BRCAness “molecular features of tumours that are BRCA1/BRCA2 deficient, referred to historically as “BRCAness”. In 40 BRCAness genes, authors included BRCA1/2 (line 74). Please clarify whether BRCAness includes BRCA1/2 and unify the term BRCAness throughout the manuscript.

2.     Unclear the take home message, 21 cancer types is a lot, any ranking?

3.     The discussion section should be expanded, what’s your interpretation of your results compared to others, and what’s new to the field?

4.     Any clinical trial using PARPi in UCEC or KIRC?

5.     Please provide rationale why you choose 40 BRCAness genes, not 39 or 41?

6.     BRCA, in this manuscript sometimes it stands for gene, sometimes for breast cancer, confused.

7.     Figure 1. A, you showed 30 GO and 4 KEGG, whether you can show more KEGG that close the -log (P value) shown in GO? B, I think a table might be better, which is easy to show the N and compare the N among different cancers. C, it is needed?

8.     Line 174-175, and Figure 2 legend, please give full names of 33 cancers abbreviations. Figure 2 is very busy, separate somatic and germline?

9.     Line 180-184 and related Figure 3. Please give rationale why you did this analysis and what’s your conclusion. I got certain sense of A and B (somatic), but not C and D (germline). The status of BRCA1/2 will affect germline variation of other BRCAness genes? Which does not make sense for me. The line is not 0.5?

10.  Figure 4. A, add tick mark on X axis. What is the inset (pie chart) on right, it was not described in text or figure legend.

11.  Figure 5. A and B, any explanation of the conflict: hypermethylation-high expression, such as FANCA; hypomethylation-low expression, such as MYC. C is very informative, but clustered into very few cancer types, due to limited data or cutoff value?

12.  Figure 6. A: genes on left=genes listed on right; cancer_types on top=caners listed in bottom? if they are same, might remove left and top lines. B, spell out OR CI. In legend, “Red: the cancer types with significantly high expression of AURKA or PTEN.”, which red you mean, as there are red cancer_types, red OR, and red P-values. C and D are interesting results, only KIRC?

13.  Table 1 broken word and sentence, please correct.

14.  Line 47, please use better known gene name, such as NBN to NBS1, BRIP to BRIP1 (FANCJ). Line 59 you wrote BRIP1.

15.  Line 50, unclear to me “the “Signature 3” of mutation signatures in cancer [20, 21]”.

16.  Please cite the correct ref, such as BRCAness, you might need the original paper use the term, Turner N er al., Nat Rev Cancer. 2004

17.  Spell out CNV

18.  Line 81, TCGA MAF file in PanCanAtlas, correct? TCGA file from PanCanAtlas.

Author Response

We have to realize it is a hot topic, and many relevant research and review papers have been published.

Reply:

Indeed, this is the case. However, most of the reported studies were limited in exploring the increased use of PARPi in BRCAness breast and ovarian cancer with no BRCA mutation. Our current study aims to expand the possibility of using PARPi for ANY cancer types with BRCAness feature.

  1. The definition of BRCAness. The first sentence in abstract “BRCAness refers to the damaged homologous recombination (HR) function due to the defects in HR-involved non-BRCA1/2 genes” Reading referred Ref 16 for BRCAness “molecular features of tumours that are BRCA1/BRCA2 deficient, referred to historically as “BRCAness”. In 40 BRCAness genes, authors included BRCA1/2 (line 74). Please clarify whether BRCAness includes BRCA1/2 and unify the term BRCAness throughout the manuscript.

Reply:

Thank you very much for your comments. As we defined in the paper, “BRCAness refers to the damaged homologous recombination (HR) function due to the defects in HR-involved non-BRCA1/2 genes”. Hence, BRCA1/2were not included in the 40 BRCAness genes (ATM, ATR, AURKA, BAP1, BARD1, BLM, BRIP1, CDK12, CHD4, CHEK1, CHEK2, EMSY, ERCC1, FANCA, FANCC, FANCD2, FANCE, FANCF, FANCI, KMT2A, MRE11A, MYC, NBN, PALB2, PARP1, PAXIP1, PLK1, PTEN, RAD50, RAD51, RAD51B, RAD51C, RAD51D, RAD52, SAMHD1, SEM1, TP53, TP53BP1, WEE1, WRN).We corrected the wrong gene list and unified the term BRCAness across the text as highlighted in the revision.

  1. Unclear the take home message, 21 cancer types is a lot, any ranking?

Reply:

We ranked the 33 cancer types based on the sum of the six BRCAneass features of somatic pathogenic variation, germline pathogenic variation, homozygous deletion, expression, and clinical prognosis. By using the sum in both BRCA cancer and OV cancer values = 1 as the cut-off, we observed that the following 21 cancer types had BRCAness features higher than BRCA cancer and OV cancer (in decreasing order): UCEC, BLCA, PAAD, LGG, SARC, LUAD, KICH, UCS, ACC, COAD, LIHC, ESCA, HNSC, READ, STAD, SKCM, LUSC, MESO, KIRC, KIRP, and PRAD (Table 1).

  1. The discussion section should be expanded, what’s your interpretation of your results compared to others, and what’s new to the field?

Reply:

Thank you very much for your suggestion. We added the relevant discussion below following your comments:

      By targeting multiple oncogenic components, synthetic lethal has shown promising potential as best exemplified by using PARPi to treat BRCA1/2 mutated breast and ovarian cancer. In our current study, we analyzed the potential of using PARPi therapy to treat other cancer types with BRCAness features. Through analyzing multiple features in 33 cancer types, our study provided the following evidence showing high similarity between BRCAness and BRCA1/2mutation in multiple cancer types:(1) Genetic variation was widely presence in BRCAness genes in multiple cancer types as represented by UCEC, BLCA, LUSC, HNSC, STAD, and COAD [44]; (2) Homozygotic variation was rare event in BRCAness genes as in BRCA1/2mutation. Similar to the embryonic lethal effects in BRCA1/2, homozygous variation in BRCAness genes was not common across different cancer types although homozygous deletions in some BRCAness genes (FANCD2, PTEN, TP53, BRCA2) could be present

[45]; (3) Promoter methylation was common in many BRCAness genes and caused alterative expression [46]. Similar to the promoter methylation occurred in BRCA1/2, promoter methylation was present in nearly half of the BRCAness genes, and their expression were silenced in many cancer types in cancer type-specific manner; (4) The defects in BRCAness genes were strong prognostics markers as BRCA1/2mutation. As shown in the BRCAness genes of CHEK2, ATM, RAD51D, EMSY, PALB2, BRIP1, ERCC1, RAD50, ATR, RAD51Cin ovarian cancer [47].

      Our study reveals that BRCAness defects is commonly present in multiple cancer types as BRCA1/2defects in breast and ovarian cancer. Therefore, it opens a possibility to further test the potential of expanding PARPi therapy from breast and ovarian cancer to more cancer types with BRCAness features.

  1. Any clinical trial using PARPi in UCEC or KIRC?

Reply:

Thank you very much for your comments. Until now, there has been many clinical trials using PARPi in UCEC or KIRC, especially KIRC as recorded ClinicalTrials.govof NIH (https://beta.clinicaltrials.gov). For examples, the clinical trial of combination of PD-1 inhibitor pembrolizumab and PARP inhibitor olaparib started in December 3, 2020 is at Phase 2 stage for the treatment of Cervical Cancer (Identifier: NCT04483544). A Phase 2 study of PARPi pamiparib (BGB-290) plus chemotherapy drug temozolomide for Hereditary Leiomyomatosis and Renal Cell Cancer (HLRCC) started on October 18, 2021 (Identifier: NCT04603365).

  1. Please provide rationale why you choose 40 BRCAness genes, not 39 or 41?

Reply:

Thank you very much for your suggestion. We identified 40 BRCAness genes from the review[35-37]. The three review papers made extensive analysis and provide solid evidence for the BRCAness features in these 40 BRCAness genes. These 40 genes are also mostly HRD-related genes.

  1. BRCA, in this manuscript sometimes it stands for gene, sometimes for breast cancer, confused.

Reply:

Thank you very much for your comments. In the revision, we replaced all “BRCA” with “BRCA1/2” if it refers to genes, and kept “BRCA” in the manuscript if it refers to breast cancer.

  1. Figure 1. A, you showed 30 GO and 4 KEGG, whether you can show more KEGG that close the -log (P value) shown in GO? B, I think a table might be better, which is easy to show the N and compare the N among different cancers. C, it is needed?

Reply:

Thank you very much for your suggestion. In Figure 1A, we only enriched 3 significant KEGG pathways using BRCA1/2and 40 BRCAness genes. For Figure 1B,  we have replaced it with a table. We deleted C from Figure 1.

  1. Line 174-175, and Figure 2 legend, please give full names of 33 cancers abbreviations. Figure 2 is very busy, separate somatic and germline?

Reply:

Thank you very much for your suggestion. We replaced the abbreviations with the full names for all 33 cancer types in Figure 2 legend. We combined somatic and germline pathogenic variants together in order to help seeing the differences between the two types of patterns.

  1. Line 180-184 and related Figure 3. Please give rationale why you did this analysis and what’s your conclusion. I got certain sense of A and B (somatic), but not C and D (germline). The status of BRCA1/2 will affect germline variation of other BRCAness genes? Which does not make sense for me. The line is not 0.5?

Reply:

Thank you very much for your suggestion. (1). Homologous recombination pathway involves multiple genes besides BRCA1/2. Further, many genes not in HR pathway can also directly or indirectly be involved in HR. In principle, defects in these non-BRCA1/2  BRCAness genes could also result in the same consequences as caused by the deficient BRCA1/2, as described as BRCAness. Hence, we performed the variation frequency correlation analysis to explore whether similar variant patterns exist between BRCAness genes and BRCA1/2mutation. We calculated variation frequency for the PLP variants in each BRCAness gene in each cancer type and tested the correlation of PLP variation frequencies between BRCAness genes and BRCA1/2 in all 33 cancer types by Pearson’s correlation coefficient analysis. (2). Our aim is not to test whether similar variant frequency pattern exists between BRCAness gene and BRCA1/2, but to confirm whether the status of BRCA1/2could affect germline variation of other BRCAness genes. (3). The sentence at the end of Figure 3 legend does not make much significance, we deleted it in the revision.

  1. Figure 4. A, add tick mark on X axis. What is the inset (pie chart) on right, it was not described in text or figure legend.

Reply:

Thank you very much for your suggestion. We added gene names on X axis in Figure 4A. In the inset on right of Figure 4A,  we displayed gene list falling into each classification. For example, the geneslist (FANCD2, BARD1, FANCC, BRCA1, RAD52, RAD51D, CDK12, PARP1, WEE1) in the blue region implies that these genes have higher homozygotic variation frequency than BRCA1in 5 to 10 cancer types.

  1. Figure 5. A and B, any explanation of the conflict: hypermethylation-high expression, such as FANCA; hypomethylation-low expression, such as MYC. C is very informative, but clustered into very few cancer types, due to limited data or cutoff value?

Reply:

Thank you very much for your suggestion. In Figure 5B, almost all BRCA1/2and BRCAness genes were not significantly up-regulated or down-regulated in nearly all cancer types except BLCA, UCEC and SARC. We considered that promoter methylation level may not be the dominant element in the regulation of BRCAness gene expression. Even though, we still found some genes significantly affected by their promoter methylation levels. For example, FANCFpromoter was hypermethylated and had decreased expression in 8 cancer types of BLCA, CESC, COAD, DLBC, ESCA, HNSC, UCEC, and UCS (Figure 5C).

  1. Figure 6. A: genes on left=genes listed on right; cancer_types on top=caners listed in bottom? if they are same, might remove left and top lines. B, spell out OR CI. In legend, “Red: the cancer types with significantly high expression of AURKA or PTEN.”, which red you mean, as there are red cancer_types, red OR, and red P-values. C and D are interesting results, only KIRC?

Reply:

Thank you very much for your suggestion. In Figure 6. A, we deleted the genes and cancers on the left and top lines. B, we revised the legend as: OR, odds ratio; CI: confidence interval; Red P-values represents a higher expression of BRCAness genes associated with worse survival, and blue P-values represents an association with better survival. C and D, by using all the altered expression patterns of all BRCAness genes, we found only KIRC patients were significantly classified into high- and low-risk groups.

  1. Table 1 broken word and sentence, please correct.

Reply:

Thank you very much for your suggestion. We have revised Table 1.

  1. Line 47, please use better known gene name, such as NBNto NBS1, BRIPto BRIP1(FANCJ). Line 59 you wrote BRIP1.

Reply:

Thank you very much for your suggestion. We replaced NBNwith NBS1, BRIPwith BRIP1across the manuscript.

  1. Line 50, unclear to me “the “Signature 3” of mutation signatures in cancer [20, 21]”.

Reply:

Thank you very much for your suggestion. “Signature 3” represents a distinct pattern of base-substitution mutations named by the following two studies:

  1. Nik-Zainal S, et al. Landscape of somatic mutations in 560 breast cancer whole-genome sequences. Nature. 534(7605):47-54, 2016;
  2. Alexandrov L, et al.Signatures of mutational processes in human cancer. Nature 500, 415–421, 2013.

  1. Please cite the correct ref, such as BRCAness, you might need the original paper use the term, Turner N er al., Nat Rev Cancer. 2004

Reply:

Thank you very much for your suggestion. We have corrected the ref. paper as #16 for BRCAness in the revision.

  1. Spell out CNV

Reply:

Thank you very much for your comments. We added the full name of CNV where it first appeared in the text.

  1. Line 81, TCGA MAF file in PanCanAtlas, correct? TCGA file from PanCanAtlas.

Reply:

Thank you very much for your suggestion. We cleared the description for the data collection details in the manuscript as following:

We collected the BRCAness genomic and clinical information from these two resources: UCSC xena (http://xena.ucsc.edu/) and PanCanAtlas (https://gdc.cancer.gov/node/905/) covering 33 cancer types [38]. The details data information are: Variation data from over 10,000 cancer patients were from TCGA MAF file in PanCanAtlas; copy number variation (CNV) data detected by Affymetrix SNP 6.0 arrays were from UCSC xena; DNA methylation data detected by Illumina HumanMethylation450 BeadChip platform were from PanCanAtlas; RNA-seq data with normalized batch effects and log2 (norm_value+1) gene expression for all 33 cancer types were from PanCanAtlas; Clinical survival data were from UCSC xena. GISTIC2 was used to identify the genomic regions with significant gain or loss [39].

Round 2

Reviewer 2 Report

I am satisfied with the revision, no further comment.